# Transfer of endogenous small RNAs between branches of scions and rootstocks in grafted sweet cherry trees

Dongyan Zhao[1,2], Gan-yuan Zhong[3], Guo-qing Song[1]*

1 Plant Biotechnology Resource and Outreach Center, Department of Horticulture, Michigan State University, East Lansing, MI, United States of America, 2 Biotechnology Center, Cornell University, Ithaca, NY, United States of America, 3 Grape Genetics Research Unit, USDA-ARS, Geneva, NY, United States of America

* songg@msu.edu

**Data Availability Statement:** All relevant data are within the manuscript and its Supporting Information files.

**Funding:** This research was supported by AgBioResearch of Michigan State University. There

## Abstract

Grafting is a well-established agricultural practice in cherry production for clonal propagation, altered plant vigor and architecture, increased tolerance to biotic and abiotic stresses, precocity, and higher yield. Mobile molecules, such as water, hormones, nutrients, DNAs, RNAs, and proteins play essential roles in rootstock-scion interactions. Small RNAs (sRNAs) are 19 to 30-nucleotides (nt) RNA molecules that are a group of mobile signals in plants. Rootstock-to-scion transfer of transgene-derived small interfering RNAs enabled virus resistance in nontransgenic sweet cherry scion. To determine whether there was long-distance scion-to-rootstock transfer of endogenous sRNAs, we compared sRNAs profiles in bud tissues of an ungrafted 'Gisela 6' rootstock, two sweet cherry 'Emperor Francis' scions as well as their 'Gisela 6' rootstocks. Over two million sRNAs were detected in each sweet cherry scion, where 21-nt sRNA (56.1% and 55.8%) being the most abundant, followed by 24-nt sRNAs (13.1% and 12.5%). Furthermore, we identified over three thousand sRNAs that were potentially transferred from the sweet cherry scions to their corresponding rootstocks. In contrast to the sRNAs in scions, among the transferred sRNAs in rootstocks, the most abundant were 24-nt sRNAs (46.3% and 34.8%) followed by 21-nt sRNAs (14.6% and 19.3%). In other words, 21-nt sRNAs had the least transferred proportion out of the total sRNAs in sources (scions) while 24-nt had the largest proportion. The transferred sRNAs were from 574 cherry transcripts, of which 350 had a match from the *Arabidopsis thaliana* standard protein set. The finding that "DNA or RNA binding activity" was enriched in the transcripts producing transferred sRNAs indicated that they may affect the biological processes of the rootstocks at different regulatory levels. Overall, the profiles of the transported sRNAs and their annotations revealed in this study facilitate a better understanding of the role of the long-distance transported sRNAs in sweet cherry rootstock-scion interactions as well as in branch-to-branch interactions in a tree.

was no additional external funding received for this study.

**Competing interests:** The authors have declared that no competing interests exist.

## Introduction

Grafting, a bud or twig as scion attached onto the lower part of another plant as rootstock, has been used widely in plant clonal propagation and crop production, especially for fruit trees [1, 2]. Its benefits include dwarfism of trees for easy harvest, increase of disease and pest resistance, and reduction of juvenility, etc. [3]. The underlying mechanisms for producing a successful graft have been studied for decades, of which how rootstocks and scions communicate with each other remains an unresolved research problem. Thus far, many metabolites, protein, mRNA, and small RNAs, including microRNAs (miRNAs) and small interfering RNAs (siRNAs) are all suggested to be potential molecular signals facilitating rootstock-scion communications [4–13].

Taking the advantage of heterografts between different species of the *Cucurbitaceae*, Bolecki and his colleagues demonstrated that at least nine proteins, including the filament-forming phloem protein 1 (PP1) and the phloem lectin PP2, were found in developing scion exudate, which corresponded to those of the respective proteins in the rootstock [7]. It was suggested that PP1 and PP2 were likely involved in transporting macromolecules within the phloem [14]. FLOWERING LOCUS T (*FT*) is a top candidate of the florigen because many reports have demonstrated that FT proteins, instead of *FT* RNAs, acted as the mobile florigenic signals mainly through short-distance transport (*e.g.*, from leaves to their adjacent buds) [15–24]. Two other reports, however, suggested that both FT proteins and *FT* RNAs could be transmitted [25, 26]. In additional, phytohormones regulated by *FT* expression may serve as important signals in long-distance transfer of floral inductive signals [27].

Similarly, transcription profile in phloem-sap of heterografts of melon stocks and pumpkin scions revealed the presence of long-distance transport of mRNAs from stocks to scions [28]. The long-distance trafficking of mRNAs *via* phloem and its impact on development of various plant organs (leaf, tuber, root, and flower) in grafted plants were enumerated in detail by Spiegelman and colleagues [29].

Small RNAs are a short stretch of RNA molecules, usually referring to those with size ranging from 19 to 30-nt. Small RNAs, including miRNAs and siRNAs, have been suggested to function as both short- and long-distance trafficking signals [3, 30–35]. Using miRNA microarray and grafts of wild type *Brassica* to mutants, miR399 and miR395 were found to be able to translocate through graft unions and function as regulating molecules in responses to stress and nutrient deficiency [36]. Another miRNA, miRNA156, has been known to be key in maintaining plant juvenile phase and mobility assays in potato heterografts suggested that it was a graft-transmissible signal that affects potato architecture and tuberization [31]. In transgrafted *Arabidopsis* mutants, transfer of a green fluorescent protein (GFP)-derived sRNAs (here 21–24 nt) as well as a substantial amount of endogenous sRNA through the graft unions were reported, and that 24-nt sRNAs directed epigenetic modifications in the recipient cells [9].

We have previously reported long-distance transfer of sRNAs generated from a hairpin-RNA transgene in transgenic 'Gisela 6' (*Prunus. cerasus × P. canescens*) rootstocks to nontransgenic sweet cherry (*P. avium* L.) scions [13]. Here, we report the findings of endogenous sRNAs transferred from sweet cherry scions to rootstocks. We showed that 21-nt sRNAs were the most abundant while 24-nt sRNAs were the second abundant sRNA species in the two sweet cherry scions. Interestingly, among the scion-to-rootstock transferred sRNAs, 24-nt sRNAs were the most abundant, which is in agreement with the role of 24-nt sRNAs acting as long-distance silencing signals. Additionally, it indicated that sRNA movement was likely a selective and somewhat controlled process. In summary, together with the previous finding of rootstock-to-scion sRNA transfer, the finding of sRNA transfer from scions to rootstocks will

help understanding the regulation of communications between rootstocks and scions and between branches, which ultimately influences the development of grafted trees.

## Materials and methods

### Grafting and sample collection

*In vitro* 'Gisela 6' shoots were rooted and grown in a greenhouse to reach about 30 cm tall. Bark graft was used to attach an individual bud of a sweet cherry (*P. avium*) 'Emperor Francis' to the trunk position of about 15 cm of a 'Gisela 6' tree (Fig 1A). Both grafted and ungrafted trees were grown in the courtyard between two greenhouses under natural light and temperature conditions in East Lansing, Michigan. The trees were irrigated and fertilized using a regular schedule to keep them healthy. To study endogenous sRNA transfer between branches in a grafted tree, we selected two 17-month old grafted 'Emperor Francis' trees in which the 'Gisela 6' branches below the graft union were retained. The selected trees were about 2 m tall by the time of sample collection in mid-October. Two bud samples, 30–50 buds per sample, from each of the two trees were collected from the branches above 1.5 m trunk positions and from the branches of the 'Gisela 6', separately. Meanwhile, 30–50 buds were collected from the branches above 1.5 m trunk positions of an ungrafted 'Gisela 6' tree. The collected buds in 2.0 ml cryotubes were frozen immediately in liquid nitrogen and then stored in a -80 ºC freezer for RNA isolation.

### Small RNA extractions and sequencing

Total RNA was isolated using a cetyltrimethylammonium bromide (CTAB) method [37]. The samples were purified using miRNeasy Mini Kit (Qiagen, Valencia, CA). Integrity of the RNA samples was assessed using the Agilent RNA 6000 Pico Kit (Agilent Technologies, Inc. Waldbronn, Germany). Small RNA libraries were constructed using the Illumina TruSeq® Small RNA Sample Prep Kit (Illumina, Inc., Hayward, CA), which were pooled and sequenced [50-bp (base pair) single end reads] using the Illumina HiSeq2500 platform at the Research Technology Support Facility of Michigan State University (East Lansing, MI).

### Transcriptome assembly and annotation

The transcriptome of sweet cherry (*P. avium* L. 'Tieton') was generated using Trinity (v20140717) [38] using mRNAseq reads downloaded from NCBI SRA (SRA Sample #: SRS671080). The representative transcripts were obtained by selecting the longest isoform of each transcript. Functional annotation was assigned by searching against the *Arabidopsis thaliana* annotation (TAIR10) and Swiss-Prot plant protein database using NCBI BLAST [39], and the Pfam (v29) [40] using HMMER (v3.1b1) [41].

### Small RNA discovery

The quality of raw Illumina reads generated above was assessed using FASTQC (v0.11.2) [42] using default parameters and raw reads were processed by removing residual adapter sequences and low-quality bases using Cutadapt (v1.8) [43]. The cleaned reads were aligned to the above-mentioned sweet cherry transcriptome using bowtie (v1.1.1) [44], only allowing alignments for reads having one reportable alignment that have no mismatches (—strata–best -m 1 -v 0). The candidate transferred small RNAs were obtained by taking the sRNAs present in grafted rootstocks (RS15 and RS19) but absent in the ungrafted rootstock (RS3). The higher confidence transferred sRNAs were obtained by requiring their presence in the transferred sRNA pools of both RS15 and RS19.

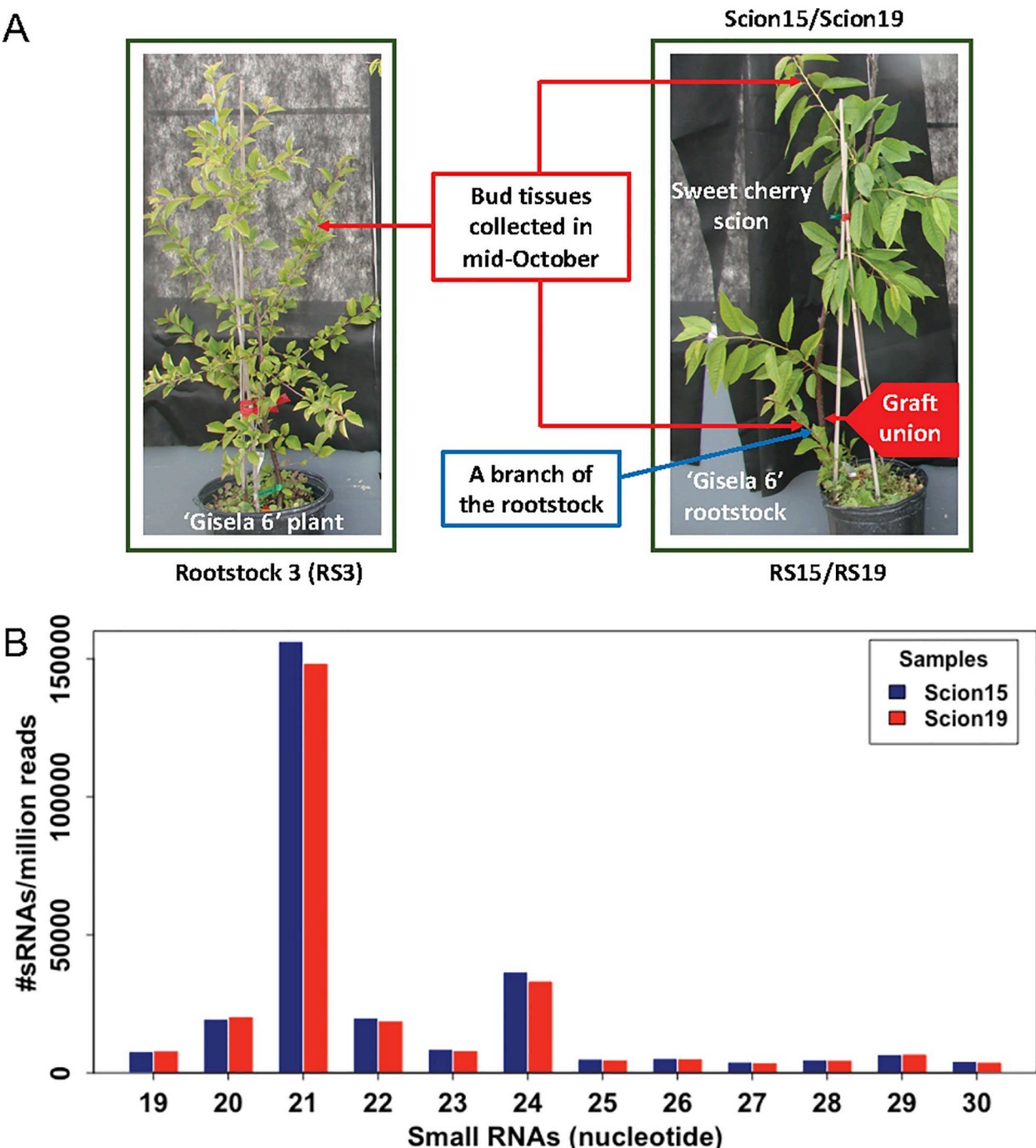

**Fig 1.** Experimental design and sample collection (**A**) and small RNA profiles in sweet cherry scions (**B**).

## Gene network analysis

We imported annotated sRNAs to Cytoscape_v3.7.2 using BiNGO to construct a gene network of the overrepresented Gene Ontology (GO) terms with selected Ontology file "GO_full" and selected organism *A. thaliana* [45, 46].

## Results and discussion

### Endogenous small RNAs in sweet cherry scions

Previously, it was reported that small RNAs were transferred from rootstock to scion, suggesting that sRNA signals may play a role in rootstock-scion communication [9, 13, 47, 48]. In this study, the rootstocks ('Gisela 6') are different species from the scions ('Emperor Francis'), which allowed the distinction of sRNAs specific from scions. 'Gisela 6' rootstock is a triploid, precocious, semi-dwarf rootstock generated from crosses between tetraploid *P. cerasus* (tart cherry) and diploid *P. canescens* [49] whereas the scions are of the species sweet cherry (*P. avium*). The nature that the rootstocks and scions are different species made it possible for us to identify transferred sRNAs that are unique for either species but with the disadvantage of the underestimation of sRNAs from highly conserved sequences shared by the two species. We used two unique grafted sweet cherry trees, in which some branches of the rootstocks were retained, to determine: 1) Whether endogenous sRNAs can be transferred in the opposite direction, from scion to rootstock; 2) Whether endogenous sRNAs can be transferred in a long distance between branches; and 3) What the potential roles of the total and transferred sRNAs are. To such end, we compared the sRNA profiles of a 'Gisela 6' rootstock (RS3) and two grafted sweet cherry 'Emperor Francis' trees (two replicates: Scion15 and Scion19) on 'Gisela 6' rootstocks (two replicates: RS15 and RS19) (Fig 1A). Immature buds were collected from branches of the 'Gisela 6' rootstocks and sweet cherry 'Emperor Francis' scions, where the distance of these buds from scion and rootstock was ~1.5 meters. Small RNA sequencing was conducted to get the pools of sRNAs. Overall, a total of 63 million 50 bp single-end reads were generated, where >96% were with Phred qualities equal to or greater than 30 (Table 1).

In order to categorize the sRNAs, the transcriptome of sweet cherry (*P. avium* L. 'Tieton') was generated using mRNAseq reads downloaded from NCBI SRA (SRA Sample #: SRS671080). The resulting transcriptome consisted of 60,946 representative transcripts, ranging from 151 to 15,645 bp in size and N50 transcript size of 1,582 bp (Table 2). The transcripts were annotated by searching for Pfam domains and aligning to the Arabidopsis and Swissprot plant protein sequences (S1 Data).

The sRNAs (19–30 nt) of the samples were obtained by aligning the reads to the representative transcriptome and then categorized according to their lengths. First, we compared sRNAs in the two sweet cherry scions. The amount of sRNAs detected were comparable between the two sweet cherry scions (S1 and S2 Tables; 2.5 million and 2.4 million sRNAs for Scion15 and Scion19, respectively), with 21-nt sRNA (56.1% and 55.8%) being the most abundant, followed by 24-nt (13.1% and 12.5%) sRNAs (Fig 1B and Table 3).

The detected sRNAs were from 17,356 transcripts in Scion15 and 16,592 transcripts in Scion19, where 11,215 and 10,837 have best matches with the *Arabidopsis* proteome for Scion15 and Scion19, respectively (S1 and S2 Tables). Among 78 unique transcripts producing over 1,000 (111 sRNAs/million reads) 21-nt and 24-nt sRNAs in Scion15, nineteen had

**Table 1. Summary of small RNA sequencing reads.**

| Sample | Pass-Filter Reads | Q-Score ≥ 30 | Average Q-Score | Yield (Gbp) |
|---|---|---|---|---|
| RS3 | 23,333,748 | 95.0% | 37.3 | 1.17 |
| RS15 | 8,507,397 | 96.5% | 37.9 | 0.43 |
| Scion15 | 10,691,794 | 96.7% | 38.0 | 0.53 |
| RS19 | 10,021,317 | 96.9% | 38.0 | 0.50 |
| Scion19 | 10,876,609 | 96.8% | 38.0 | 0.54 |
| **Total** | **63,430,865** | | | |

**Table 2. Metrics of the representative transcriptome of sweet cherry (*Prunus avium* L. 'Tieton').**

| Metrics | Value |
|---|---:|
| Number of representative transcripts | 60,946 |
| Longest transcript | 15,645 bp |
| Shortest transcript | 151 bp |
| N50 transcript size | 1,582 bp |
| Average transcript size | 729 bp |

matches in *Arabidopsis* proteome. Gene Ontology (GO) analysis revealed that "transporter activity" in "molecular function" and "nucleus" in "cellular compartment" were the most prevalent in the nineteen transcripts. In the analysis of the overrepresented GO terms, 29 were identified in "biological process", three were in "molecular function", and no overrepresented GO terms were found in "cellular component". Of the 29 overrepresented GO terms in "biological process", 16 were annotated as "negative regulation of . . .", which were all related to "negative regulation of biological process" in addition to the other four overrepresented GO terms including "regulation of cell communication", "regulation of signaling pathway", "regulation of abscisic-acid mediated signaling pathway", and "regulation of response to stimulus" (Fig 2A). Three overrepresented GO terms in "molecular function" included "DNA helicase activity", "ATPase activity", and "DNA helicase activity" (Fig 2B). Similar GO results were observed for Scion19 (S1 Fig). These overrepresented GO terms revealed the potential roles of the sRNAs produced in bud tissues collected in mid-October in Michigan before the trees became dormant.

## Transfer of endogenous small RNAs from scion to rootstock

The transferred sRNAs were obtained using a grafted vs. ungrafted rootstock subtraction method (Fig 3). This resulted in 3,614 sRNAs from 2,169 transcripts for graft Scion15-to-RS15 and 3,225 sRNAs from 2,455 transcripts for Scion19-to-RS19, which were candidate sRNAs transferred from sweet cherry scions to rootstocks (Fig 4, Table 3). Overall, the number of transferred sRNAs per transcript is low, ranging from 1 to 29. In contrast to the sRNA profiles

**Table 3. The number and fraction (%) of sRNAs in sweet cherry scions and potential scion-to-rootstock transferred sRNAs.**

| sRNA species | Scion15 (%) | | Scion19 (%) | | RS15_specific (RS15 vs. RS3) (%) | | RS19_specific (RS19 vs. RS3) (%) | |
|---|---:|---|---:|---|---:|---|---:|---|
| 19-nt | 70,186 | (2.80) | 71,087 | (3.01) | 133 | (3.68) | 163 | (5.05) |
| 20-nt | 175,812 | (7.02) | 181,400 | (7.67) | 165 | (4.57) | 188 | (5.83) |
| 21-nt | 1,405,289 | (56.10) | 1,318,390 | (55.78) | 529 | (14.64) | 622 | (19.29) |
| 22-nt | 179,587 | (7.17) | 167,289 | (7.08) | 329 | (9.10) | 334 | (10.36) |
| 23-nt | 76,850 | (3.07) | 71,610 | (3.03) | 253 | (7.00) | 200 | (6.20) |
| 24-nt | 328,641 | (13.12) | 296,071 | (12.53) | 1,674 | (46.32) | 1,123 | (34.82) |
| 25-nt | 45,122 | (1.80) | 42,368 | (1.79) | 150 | (4.15) | 128 | (3.97) |
| 26-nt | 47,801 | (1.91) | 45,623 | (1.93) | 84 | (2.32) | 133 | (4.12) |
| 27-nt | 35,566 | (1.42) | 32,873 | (1.39) | 74 | (2.05) | 77 | (2.39) |
| 28-nt | 42,712 | (1.70) | 40,730 | (1.72) | 92 | (2.55) | 83 | (2.57) |
| 29-nt | 59,896 | (2.39) | 61,128 | (2.59) | 70 | (1.94) | 85 | (2.64) |
| 30-nt | 37,731 | (1.51) | 35,088 | (1.48) | 61 | (1.69) | 89 | (2.76) |
| **Total** | 2,505,193 | | 2,363,657 | | 3,614 | | 3,225 | |
| **Transcripts** | 17,356 | | 16,592 | | 2,169 | | 2,455 | |

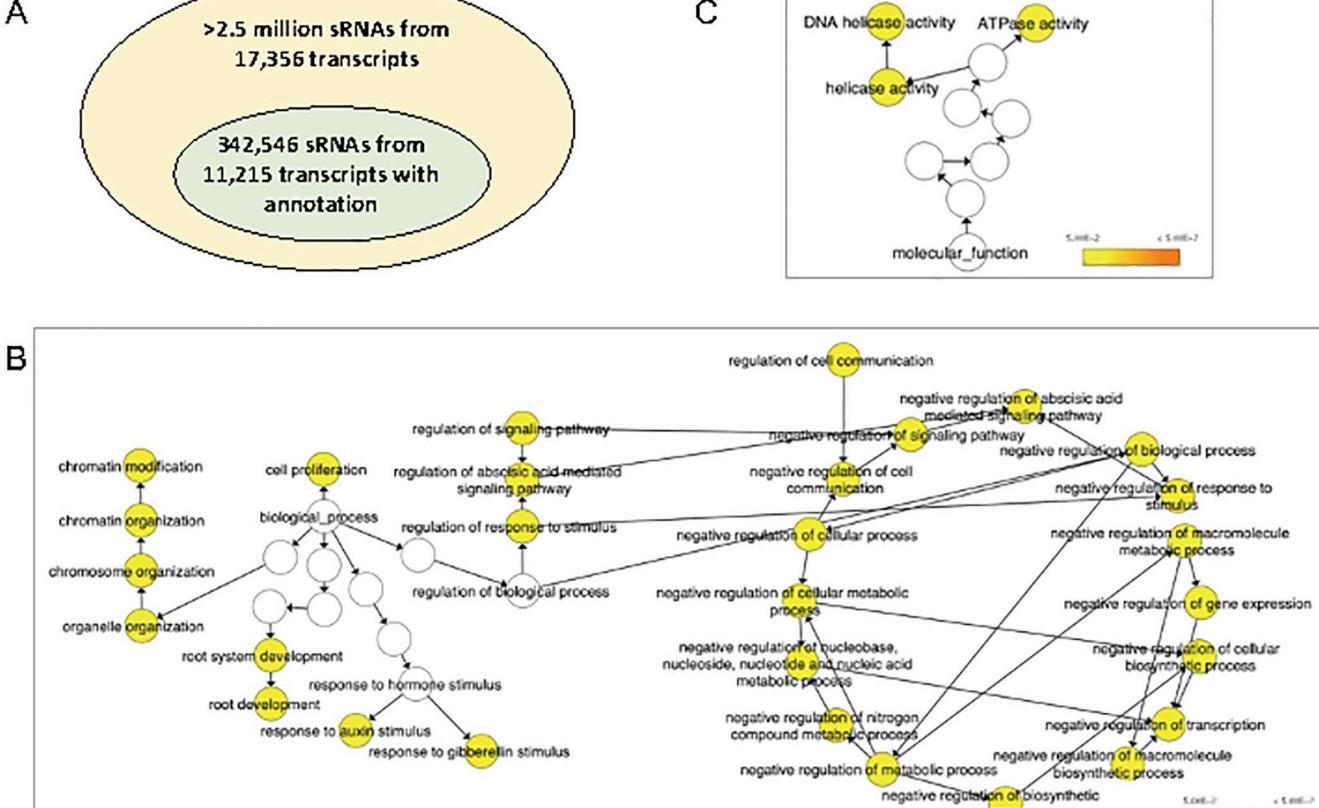

**Fig 2.** Summary of sRNAs (A) and gene networks of overrepresented sRNAs in bud tissues of sweet cherry scion 15 (Scion15) 'Emperor Francis' grafted on a 'Gisela 6' rootstock (RS15). The ontology file of GO_FULL in BiNGO and *A. thaliana* annotation were used as the references to identify overexpressed GO terms ($P < 0.05$). Bubble color indicates the *P*-value. Overrepresented sRNAs in "Biological process" (B) and "Molecular function" (C). No overrepresented GO terms are present in "Cellular component".

in scions, 24-nt (46.3% and 34.8% for Scion15-to-RS15 and Scion19-to-RS19, respectively) sRNAs are the most abundant followed by 21-nt (14.6% and 19.3% for Scion15-to-RS15 and Scion19-to-RS19, respectively) sRNAs in the scion-to-rootstock transferred sRNAs (Fig 4). To be more confident, we looked further into the common sRNAs in the two scion-to-rootstock transferred sRNA pools. This resulted in a total of 1,991 sRNAs from 574 transcripts. Similarly, 24-nt (38.0%) and 21-nt (18.9%) sRNAs were the two most abundant species. Among the small RNAs, functions of 21-nt and 24-nt species are the most well-studied, both having critical silencing functions within a given organism [9, 36]. The overrepresentation of 24-nt sRNAs is in agreement with previous research findings that 24-nt sRNAs function in long-range silencing [50]. It was suggested that 24-nt sRNA could initiate DNA methylation of the recipient cells, transcriptional regulation, and epigenetic silencing [34].

Previous research suggested sRNA movements happen through a bulk flow process in the phloem [36]. To determine whether there was any correlation between sRNAs in scions and rootstocks, Pearson correlation coefficient was calculated between sRNAs in scions (source) and scion-to-rootstock (sink) transferred sRNAs. A technically positive (R = 0.35, $p = 0.26$ for Scion15 and Scion15-to-RS15 transferred sRNAs) and a moderate positive (R = 0.53, $p = 0.07$ for Scion19 and Scion19-to-RS19 transferred sRNAs) correlation were detected, albeit, neither was statistically significant at *p*-value of 0.05. While for most sRNA species, the proportions of

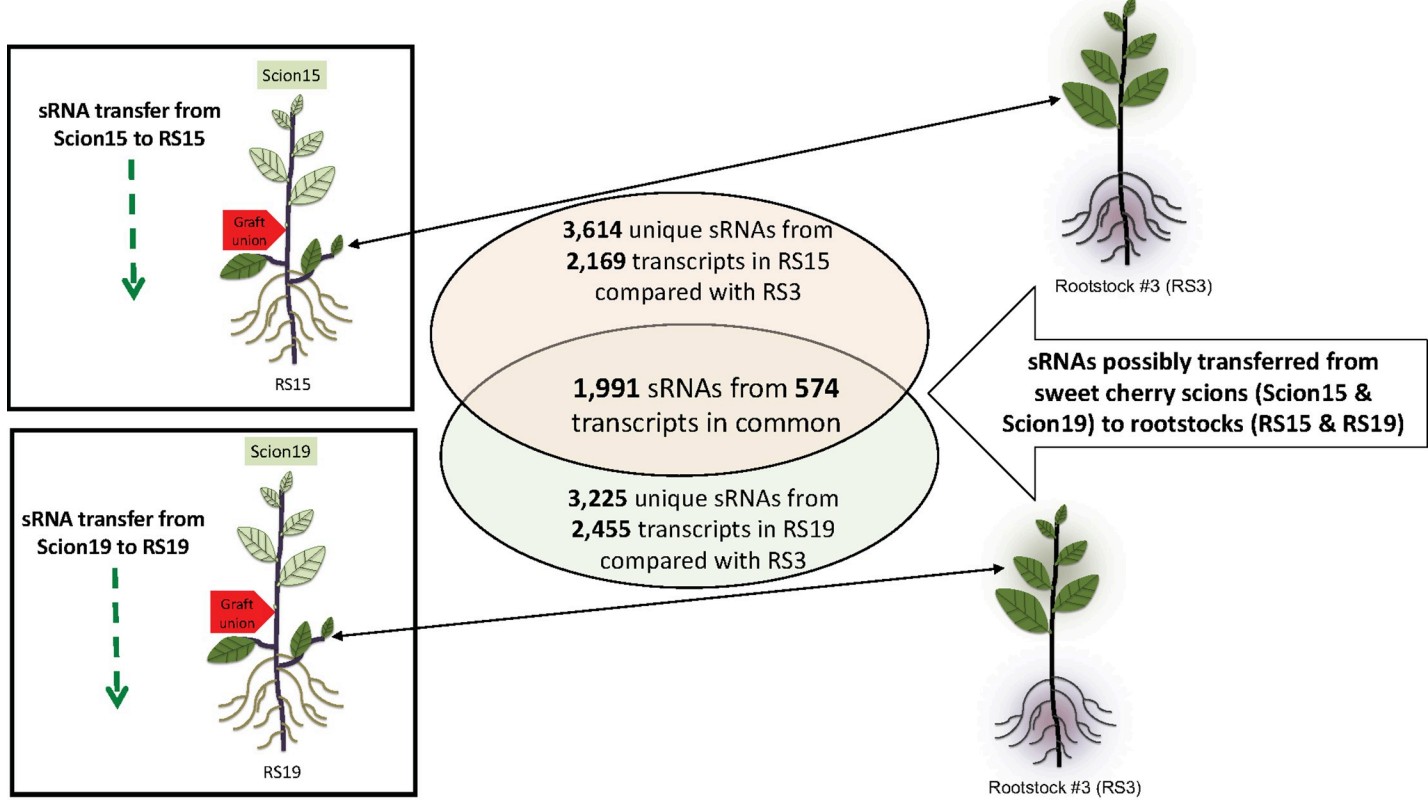

**Fig 3. A diagram showing the pipeline of determining scion-to-rootstock transferred sRNAs.**

transferred sRNAs were relatively similar (1–2‰ of sRNAs in scions), it was particularly intriguing that the transferred 21-nt sRNAs were only 0.38‰ (Scion15-to-RS15) and 0.47‰ (Scion19-to-RS19) of the total sRNAs detected in the source (scions), which were the lowest among all sRNA species. On the contrary, the proportion of transferred 24-nt sRNAs accounted for 5.09‰ (Scion15-to-RS15) and 3.79‰ (Scion19-to-RS19) of the sRNAs in the source, which were 14 and 8-fold higher than that of 21-nt sRNAs. The uneven transfer of different sRNA species suggested sRNA transfer may be selective and somewhat controlled. More in-depth genetic and genomic studies are required to further demystify this question.

## Transferred small RNAs belong to transcripts involved in binding and hydrolase activities

Among the 574 transcripts with transferred sRNAs common in the two replicates (S3 Table), 350 were annotated with best protein matches in Arabidopsis. To determine the potential function of these sRNAs, gene ontology analysis was done. Within the biological process category, transcripts with "DNA or RNA binding activity" ($\chi^2$ = 3.263, $p$ = 0.07) and "hydrolase activity" ($\chi^2$ = 1.133, $p$ = 0.29) were enriched, but not statistically significant. Proteins with "DNA or RNA binding activity" regulate many cellular processes, including transcription, mRNA processing, translation, gene silencing [51]. Therefore, the transferred sRNAs of these transcripts may affect the biological processes of the recipient plants (rootstocks) at different levels.

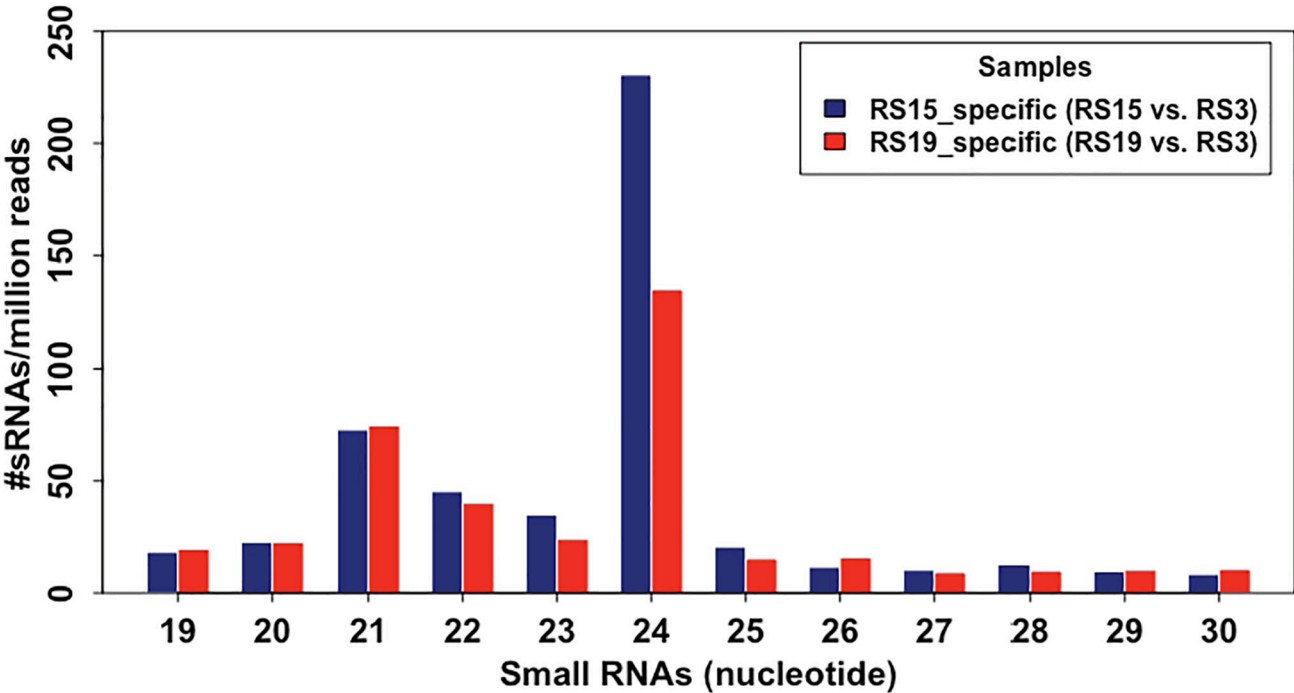

**Fig 4. Profiles of putative sRNAs transferred from sweet cherry scions to rootstocks.** RS15_specific and RS19_specific are sRNAs found in the grafted rootstocks, RS15 and RS19, but absent in the ungrafted.

## Summary

Although the transfer of sRNAs does not indicate a physiological role by itself, it is highly likely that small RNAs transferred from scion to rootstock may affect the development of the rootstock as was shown that rootstock-to-scion transferred sRNAs enabled virus resistance in scion. Collectively, sRNA transfer can happen in either direction between rootstocks and scions, likely being involved in inter-communications between scions and rootstocks.

## Supporting information

**S1 Fig.** Summary of sRNAs (A) and gene networks of overrepresented sRNAs in bud tissues of sweet cherry scion 19 (Scion19) 'Emperor Francis' grafted on a 'Gisela 6' rootstock (RS19). The ontology file of GO_FULL in BiNGO and *A. thaliana* annotation were used as the references to identify overexpressed GO terms ($P < 0.05$). Bubble color indicates the *P*-value. Overrepresented sRNAs in "Biological process" (B) and "Molecular function" (C). No overrepresented GO terms are present in "Cellular component".
(TIFF)

**S1 Table. Small RNAs in sweet cherry Scion15.**
(XLSX)

**S2 Table. Small RNAs in sweet cherry Scion19.**
(XLSX)

**S3 Table. Potential scion-to-rootstock transferred sRNAs in common between rootstocks, RS15 and RS19.**
(XLSX)

**S1 Data. The transcriptome of sweet cherry (*P. avium* L. 'Tieton').**
(FA)

## Author Contributions

**Conceptualization:** Gan-yuan Zhong, Guo-qing Song.

**Data curation:** Dongyan Zhao, Gan-yuan Zhong, Guo-qing Song.

**Formal analysis:** Dongyan Zhao.

**Funding acquisition:** Guo-qing Song.

**Investigation:** Dongyan Zhao, Guo-qing Song.

**Methodology:** Guo-qing Song.

**Project administration:** Guo-qing Song.

**Resources:** Guo-qing Song.

**Supervision:** Guo-qing Song.

**Writing – original draft:** Dongyan Zhao.

**Writing – review & editing:** Gan-yuan Zhong, Guo-qing Song.

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
