## [Decision Letter · Decision Letter 0]

7 May 2020

PONE-D-20-11020

Transfer of endogenous small RNAs between branches of scions and rootstocks in grafted sweet cherry trees

PLOS ONE

Dear Dr Song,

Thank you for submitting your manuscript to PLOS ONE. After careful consideration, we feel that it has merit but does not fully meet PLOS ONE’s publication criteria as it currently stands. Therefore, we invite you to submit a revised version of the manuscript that addresses the points raised during the review process.

We would appreciate receiving your revised manuscript by Jun 21 2020 11:59PM. To enhance the reproducibility of your results, we recommend that if applicable you deposit your laboratory protocols in protocols.io, where a protocol can be assigned its own identifier (DOI) such that it can be cited independently in the future. For instructions see: http://journals.plos.org/plosone/s/submission-guidelines#loc-laboratory-protocols

We look forward to receiving your revised manuscript.

Kind regards,

Yuan Huang

Academic Editor

PLOS ONE

Journal Requirements:

Reviewers' comments:

Reviewer's Responses to Questions

**Comments to the Author**

1. Is the manuscript technically sound, and do the data support the conclusions?

Reviewer #1: Yes

Reviewer #2: Partly

2. Has the statistical analysis been performed appropriately and rigorously? 

Reviewer #1: No

Reviewer #2: I Don't Know

3. Have the authors made all data underlying the findings in their manuscript fully available?

Reviewer #1: Yes

Reviewer #2: No

4. Is the manuscript presented in an intelligible fashion and written in standard English?

Reviewer #1: No

Reviewer #2: Yes

5. Review Comments to the Author

Reviewer #1: In the manuscript entitled “Transfer of endogenous small RNAs between branches of scions and rootstocks in grafted sweet cherry trees”, the author reported that endogenous small RNAs of grafted sweet cherry trees can be transferred from scion to rootstock. GO analysis was used to demonstrate communication functions of long-distance transported sRNAs. However, there are still a lot of queries need to be addressed.

Major

1. Author mentioned that “small RNAs between branches of scions and rootstocks”, but actually the samples were buds in the branches but not whole branches. What was the agricultural meaning of sRNAs transport between buds?

2. As we know that fruit tree is a high heterozygous species. I am wondering, are those two ‘Gisela 6’ stocks grafted with scion15/19 and ungrafted stock the same ‘Gisela 6’ propagated by asexual reproduction? If yes, I am again wondering how to distinguish either SNPs or mobile sRNA between ‘Gisela 6’ and scions by sequencing? More detailed protocol required to describe.

3. The direction of RNA signals was reported mobile from source-to-sink, in designed experiment of Figure 1A, as main source tissue, scion15/19 derived sRNA absolutely will move to sink ‘Gisela 6’ following a major flow. So in your opinion what decide the moving direction of sRNA?

4. I think the logic of lines 169 to 186 is unclear. Perhaps you should use a Venn diagram to explain how many sRNAs have been identified and how many have been annotated, and then talk about specific functional annotations.

5. More biological evidences for example transient RNA silencing experiment need to further prove the conclusions.

Minor

1. “21” in line 165 should be replaced with “24”

2. The word "value" in Table 1 should be replaced with more specific unit, for example “reads”.

3. Please add the counts and proportions of the transferred sRNAs in Figure 3 to make the picture information more abundant. Another problem in Figure 3 is that the three blue arrows are too random to be understood.

Reviewer #2: Manuscript entitled “Transfer of endogenous small RNAs between branches of scions and rootstocks in grafted sweet cherry trees” by Zhao et al. was focused on comparison of sRNAs profiles in bud tissues of an ungrafted ‘Gisela 6’ rootstock, two sweet cherry ‘Emperor Francis’ scions as well as their ‘Gisela 6’ rootstocks to determine whether there was long-distance scion-to rootstock transfer of endogenous sRNAs. This was an interesting scientific problem, and the manuscript was well writing and clear demonstration. However, in my options, this work was primarily and other questions as follows limited acceptance for publication in this journal.

1. Authors believed some sRNAs involvement in transport between rootstock and scion, which based on the data of bioinformatics analysis, but not biochemical analysis, so the results should check again by PCR.

2. Authors analyzed function of target gene of sRNAs from scion by bioinformatics, perhaps the target gene of sRNAs from scion was different from rootstock, so the GO work were insufficiency of evidence.

3. Material cultivation environmental parameters such as temperature, humidity, illumination et al. and grafting process and method were suggested to be supplied in the paper.

4. In fact, I cannot understand how to determinate candidate transferred small RNAs? Because I believe grafting maybe induced sRNAs expression but not rootstock or scion.

6. PLOS authors have the option to publish the peer review history of their article (what does this mean?). If published, this will include your full peer review and any attached files.

Reviewer #1: No

Reviewer #2: No

---

## [Author Response · Author response to Decision Letter 0]

8 Jun 2020

Reviewer #1: In the manuscript entitled “Transfer of endogenous small RNAs between branches of scions and rootstocks in grafted sweet cherry trees”, the author reported that endogenous small RNAs of grafted sweet cherry trees can be transferred from scion to rootstock. GO analysis was used to demonstrate communication functions of long-distance transported sRNAs. However, there are still a lot of queries need to be addressed.

1. Author mentioned that “small RNAs between branches of scions and rootstocks”, but actually the samples were buds in the branches but not whole branches. What was the agricultural meaning of sRNAs transport between buds?

Response: Yes, the more precise expression should be “small RNAs in buds of branches of scions and rootstocks”. The importance of studying the sRNAs in buds is in multiple folds:

a. buds are one of the most critical receptor organs of environmental and internal signals, which ultimately determine when to break buds and flower. This is agriculturally important because it’s directly related to fruit production.

b. secondly, we selected buds also because they are the distal parts of the plant with the distance between buds of scions and rootstocks the longest.

2. As we know that fruit tree is a high heterozygous species. I am wondering, are those two ‘Gisela 6’ stocks grafted with scion15/19 and ungrafted stock the same ‘Gisela 6’ propagated by asexual reproduction? If yes, I am again wondering how to distinguish either SNPs or mobile sRNA between ‘Gisela 6’ and scions by sequencing? More detailed protocol required to describe.

Response: The ‘Gisela 6’ stocks were clonally propagated. ‘Gisela 6’ was a hybrid between a tetraploid tart cherry (Prunus cerasus) and a diploid hoary cherry (Prunus canescens), both are different species from the sweet cherry (Prunus avium) scions. We acknowledge that only species-specific sRNAs were recovered in the transferred sRNA pools and those from highly conserved genes between the stocks and scions were left unaccounted, which resulted an underestimation of total transferred sRNAs. We added some texts to better express this situation. 

“‘Gisela 6’ rootstock is a triploid, precocious, semi-dwarf rootstock generated from crosses between tetraploid P. cerasus (tart cherry) and diploid P. canescens (Gutzwiler and Lang, 2001) whereas the scions are of the species sweet cherry (P. avium). The nature that the rootstocks and scions are different species made it possible for us to identify transferred sRNAs that are unique for either species but with the disadvantage of the underestimation of sRNAs from highly conserved sequences shared by the two species.”

3. The direction of RNA signals was reported mobile from source-to-sink, in designed experiment of Figure 1A, as main source tissue, scion15/19 derived sRNA absolutely will move to sink ‘Gisela 6’ following a major flow. So in your opinion what decide the moving direction of sRNA?

Response: This is a very good and hard question. Let me begin with clarifying this: sRNA transfer is bi-directional. We demonstrated the rootstock-to-scion sRNA transfer in a previous study (Zhao and Song, 2014) AND here scion-to-rootstock sRNA transfer. The “source and sink” were defined based on which direction one examined, so rootstock can be “source” when we look at sRNA transfer from rootstock to scion and “sink” when we look at sRNA transfer from scion to rootstock.

As to “what decide the moving direction of sRNAs”, there are good reviews on this topic, for example, Melnyk et al., 2011. In a few words, several mechanisms were proposed, including passive movement of sRNAs in phloem and possible selective movement for which the mechanism is still unclear. 

4. I think the logic of lines 169 to 186 is unclear. Perhaps you should use a Venn diagram to explain how many sRNAs have been identified and how many have been annotated, and then talk about specific functional annotations.

Response: Thanks for the suggestion. We added a diagram showing this part of the result (See Figure 2A for replicate Scion15 and Supplementary Figure S1A for replicate Scion19).

5. More biological evidences for example transient RNA silencing experiment need to further prove the conclusions.

Response: We completely agree with you on this, which is why we are cautious about drawing concrete implications from the sRNA results in this study. While empirical evidence is needed, it will be the focus of future work.

Minor

1. “21” in line 165 should be replaced with “24”

Response: Thanks. We corrected it.

2. The word "value" in Table 1 should be replaced with more specific unit, for example “reads”.

Response: Thanks for pointing it out. Since there are two types of values in this column, we added the unit for all of them to make it easy to read and understand.

3. Please add the counts and proportions of the transferred sRNAs in Figure 3 to make the picture information more abundant. Another problem in Figure 3 is that the three blue arrows are too random to be understood.

Response: Very good suggestion. We modified Figure 3 accordingly.

Reviewer #2: Manuscript entitled “Transfer of endogenous small RNAs between branches of scions and rootstocks in grafted sweet cherry trees” by Zhao et al. was focused on comparison of sRNAs profiles in bud tissues of an ungrafted ‘Gisela 6’ rootstock, two sweet cherry ‘Emperor Francis’ scions as well as their ‘Gisela 6’ rootstocks to determine whether there was long-distance scion-to rootstock transfer of endogenous sRNAs. This was an interesting scientific problem, and the manuscript was well writing and clear demonstration. However, in my options, this work was primarily and other questions as follows limited acceptance for publication in this journal.

1. Authors believed some sRNAs involvement in transport between rootstock and scion, which based on the data of bioinformatics analysis, but not biochemical analysis, so the results should check again by PCR.

Response: Thanks for the suggestion. You are right and biochemical analysis will be the focus of other studies and is not an area of our expertise. For our previous work (Zhao and Song, 2014), where we investigated the rootstock-to-scion transfer of sRNAs generated from a transgene in rootstock, we did qRT-PCR as well, which turned out to be less sensitive compared to sRNA sequencing (high-throughput sequencing following small RNA extraction). Therefore, we chose sRNA sequencing for the study presented here. Moreover, considering the fast and wide adoption of high-throughput sequencing for studying sRNA, the current analysis should suffice our aim in this study, which is to confirm the scion-to-rootstock transfer of sRNA.

2. Authors analyzed function of target gene of sRNAs from scion by bioinformatics, perhaps the target gene of sRNAs from scion was different from rootstock, so the GO work were insufficiency of evidence.

Response: We agree it’s a valid concern. It is possible the target genes, especially the cascade of sRNA silencing might be different between scions and rootstocks. This comes again down to the biochemical analysis as mentioned in the last comment, without which only sequence homology can be used to deduce the potential function of sRNAs. 

3. Material cultivation environmental parameters such as temperature, humidity, illumination et al. and grafting process and method were suggested to be supplied in the paper.

Response: We added the information in lines 101-104.

4. In fact, I cannot understand how to determinate candidate transferred small RNAs? Because I believe grafting maybe induced sRNAs expression but not rootstock or scion.

Response: We agree the sRNA pool is usually a result of multiple factors, including grafting and other abiotic and biotic stimuli. However, the sRNA profiling was done two-years after the grafting, when the impact of grafting (specifically, the mechanistic lesion) subsided. In addition, the fact that the rootstocks (Prunus cerasus x Prunus canescens) and scions (Prunus avium) are different species enabled the profiling of transferred sRNAs.

---

## [Decision Letter · Decision Letter 1]

18 Jun 2020

PONE-D-20-11020R1

Transfer of endogenous small RNAs between branches of scions and rootstocks in grafted sweet cherry trees

PLOS ONE

Dear Dr. Song,

Thank you for submitting your manuscript to PLOS ONE. After careful consideration, we feel that it has merit but does not fully meet PLOS ONE’s publication criteria as it currently stands. Therefore, we invite you to submit a revised version of the manuscript that addresses the points raised during the review process.

We look forward to receiving your revised manuscript.

Kind regards,

Yuan Huang

Academic Editor

PLOS ONE

Reviewers' comments:

Reviewer's Responses to Questions

**Comments to the Author**

1. If the authors have adequately addressed your comments raised in a previous round of review and you feel that this manuscript is now acceptable for publication, you may indicate that here to bypass the “Comments to the Author” section, enter your conflict of interest statement in the “Confidential to Editor” section, and submit your "Accept" recommendation.

Reviewer #2: All comments have been addressed

Reviewer #3: All comments have been addressed

2. Is the manuscript technically sound, and do the data support the conclusions?

Reviewer #2: Yes

Reviewer #3: Partly

3. Has the statistical analysis been performed appropriately and rigorously? 

Reviewer #2: Yes

Reviewer #3: N/A

4. Have the authors made all data underlying the findings in their manuscript fully available?

Reviewer #2: Yes

Reviewer #3: Yes

5. Is the manuscript presented in an intelligible fashion and written in standard English?

Reviewer #2: Yes

Reviewer #3: Yes

6. Review Comments to the Author

Reviewer #2: (No Response)

Reviewer #3: In this paper, the small RNA differences among different scions and rootstocks in grafted sweet cherry trees were analyzed, and a batch of endogenous sRNAs for potential long-distance scion-to-rootstock transfer was screened. This study provided references for sRNA function research in the biological processes of the recipient plants.

Overall, the author has replied the questions raised by Reviewer #1 and Reviewer #2, and the manuscript has been modified accordingly. However, due to the understanding of the role of the long-distance transported sRNAs in sweet cherry was all based on sRNAs profiles, It is suggested that the information about sequencing depth and quality of the original data should be provided in detail to prove the reliability of the relevant conclusions.

7. PLOS authors have the option to publish the peer review history of their article (what does this mean?). If published, this will include your full peer review and any attached files.

Reviewer #2: No

Reviewer #3: No

---

## [Author Response · Author response to Decision Letter 1]

2 Jul 2020

Reviewer #3: In this paper, the small RNA differences among different scions and rootstocks in grafted sweet cherry trees were analyzed, and a batch of endogenous sRNAs for potential long-distance scion-to-rootstock transfer was screened. This study provided references for sRNA function research in the biological processes of the recipient plants.

Overall, the author has replied the questions raised by Reviewer #1 and Reviewer #2, and the manuscript has been modified accordingly. However, due to the understanding of the role of the long-distance transported sRNAs in sweet cherry was all based on sRNAs profiles, It is suggested that the information about sequencing depth and quality of the original data should be provided in detail to prove the reliability of the relevant conclusions.

Response: Thanks for your suggestion. We added Table 1 (the summary of small RNA sequencing reads) and mentioned the read information in the manuscript as follows:

“Overall, a total of 63 million 50 bp single-end reads were generated, where >96% were with Phred qualities equal to or greater than 30 (Table 1).”

---

## [Editor Report · Decision Letter 2]

7 Jul 2020

Transfer of endogenous small RNAs between branches of scions and rootstocks in grafted sweet cherry trees

PONE-D-20-11020R2

Dear Dr. Song,

We’re pleased to inform you that your manuscript has been judged scientifically suitable for publication and will be formally accepted for publication once it meets all outstanding technical requirements.

Kind regards,

Yuan Huang

Academic Editor

PLOS ONE
---

## [Editor Report · Acceptance letter]

10 Jul 2020

PONE-D-20-11020R2 

Transfer of endogenous small RNAs between branches of scions and rootstocks in grafted sweet cherry trees 

Dear Dr. Song:

I'm pleased to inform you that your manuscript has been deemed suitable for publication in PLOS ONE. Congratulations! Your manuscript is now with our production department. 

Kind regards, 

on behalf of

Dr. Yuan Huang 

Academic Editor

PLOS ONE